COVID-19 simulation study—the effect of strict non-pharmaceutical interventions (NPIs) on controlling the spread of COVID-19

Alzu’bi Amal 1 aazoubi9@just.edu.jo
Abu Alasal Sanaa 2
http://orcid.org/0000-0003-4504-4472 Kheirallah Khalid A. 3
Watzlaf Valerie 4
1 Department of Computer Information Systems, Jordan University of Science and Technology , Irbid , Jordan
2 Department of Computer Science, Jordan University of Science and Technology , Irbid , Jordan
3 Department of Public Health, Jordan University of Science and Technology , Irbid , Jordan
4 Department of Health Information Management, University of Pittsburgh , Pittsburgh, PA , USA
Palazón-Bru Antonio
Electronic publication date: 2021 Mar 30
Publication date: 2021
Volume: 9
Electronic Location ID: e11172
Received 2020 Nov 15; Accepted 2021 Mar 7
Copyright: © 2021 Alzu’bi et al.
Copyright year: 2021
Copyright holder: Alzu’bi et al.
License: This is an open access article distributed under the terms of the Creative Commons Attribution License, which permits unrestricted use, distribution, reproduction and adaptation in any medium and for any purpose provided that it is properly attributed. For attribution, the original author(s), title, publication source (PeerJ) and either DOI or URL of the article must be cited.
License URL: https://creativecommons.org/licenses/by/4.0/

Keywords: COVID-19, SARS-COV-2, Agent-based modeling, Epidemic, NPIs

Funding: Jordan University of Science and Technology This project is funded by the Jordan University of Science and Technology. The funders had no role in study design, data collection and analysis, decision to publish, or preparation of the manuscript.

==============================
Background

From the beginning of 2020, COVID-19 infection has changed our lives in many aspects and introduced limitations in the way people interact and communicate. In this paper, we are evaluating the effect of non-pharmaceutical interventions (NPI) in limiting the spread of the Severe Acute Respiratory Syndrome Coronavirus 2 pandemic during a wedding ceremony from Irbid, Northern Jordan. Agent-based modeling was used in a real wedding event that occurred at the beginning of the spread of the pandemic in Jordan. Two infected nationals of Jordan, who arrived in Jordan about a week before the event, initiated the spread of the pandemic within the contact community.

Methods

In this work, a strict national NPI that the government implemented is developed by using an abstract model with certain characteristics similar to the Jordanian community. Thus, the Jordanian community is represented in terms of ages, occupations, and population movements. After that, the extent of the impact of the NPI measures on the local community is measured.

Results

We observed the deterioration of the state of society while the epidemic is spreading among individuals in the absence of preventive measures. Also, the results show that the herd immunity case was an epidemic, with a high level of spread among the community with 918 cases during a short interval of time. On the other hand, the preventive measures scenario shows a totally controlled spread with only 74 cases applied on the same interval of time. Furthermore, a convergence in the actual results of the real system with the hypothetical system were detected in the case of applying the strict NPI measures. Finally, strict NPI at the community level following social gatherings seem to be effective measures to control the spread of the COVID- 19 pandemic.

Introduction

At the end of 2019, Severe Acute Respiratory Syndrome Coronavirus 2 (SARS-CoV-2) evolved as a new pandemic with characteristics similar to Severe Acute Respiratory Syndrome Coronavirus (SARS-CoV) and Middle East Respiratory Syndrome Coronavirus. The World Health Organization (WHO) identified COVID-19, the disease caused by SARS-COV-2, as a pandemic (WHO, 2020). In mid-December 2020, the number of infections reached nearly 71 million confirmed cases, more than 1.5 million deaths, and around 50 million recovered persons (in Jordan 257,275 infected, 3,335 deaths and 213,244 recovered) (WorldMeters, 2020). Worldwide, there has been a focus on research about the transmission methods and the effectiveness of governmental non-pharmaceutical intervention (NPI) measures to control the spread of the pandemic (Kl^oh et al., 2020).

The main objective of these measures is to minimize mortality rates and to control the required healthcare services while preserving and considering the economic impacts. The initial response in the first weeks of the spread of the disease was reported as “very crucial” since it is very critical to influence the course of national epidemics (Cuevas, 2020). Jordan, a developing country in the Eastern Mediterranean Region, was not among the first countries in which the COVID-19 hit. As soon as the Ministry of Health (JMH, 2020) reported initial cases mid-March, the country imposed strict NPI measures that contributed to limiting the spread of the disease (WorldMeters, 2020).

These measures included suspension of schools, universities and governmental services, and closing all religious institutions, besides imposing a curfew for security and management (Kheirallah et al., 2020). Additionally, financial fines were imposed and an arrest warrant was issued to all people who broke the ban for security and management. Research has been directed towards the effectiveness of these measures (Kheirallah et al., 2020; Jordan Media Institute, 2020). It is known that waiting until the end of the epidemic may lead to high morbidity and mortality (Sun et al., 2020), and therefore, there is an urgent need to simulate these procedures in a virtual environment.

One of the most important models that applies this type of simulation is called the Susceptible, Infected and Recovered model (SIR) (Skvortsov et al., 2007), which is a mathematical model established in 1927 as one of the foundations of epidemiology and modeling of how infectious diseases can potentially spread within communities (Kermack & McKendrick, 1991). However, such models are limited to simple societies and cannot perfectly represent the characteristics, factors and movements of the population, in general, or the population subgroups (Chang et al., 2020).

One solution of this limitation is the simulation using agent-based modeling, which can represent the characteristics of the contents of the system in a detailed manner by defining a set of factors and characteristics (Currie et al., 2020). Agent-base modeling is a method of simulating a closed environment of a real-time (organized system) in an abstract representation (Tuomisto et al., 2020; Alzu’bi, Abu Alasal & Watzlaf, 2020). The main aim of this simulation is to study the behavior of the system’s agents based on predetermined rules. SIR model has been adopted by the agent-based models and has proven effectiveness in many fields (Alzu’bi, Abu Alasal & Watzlaf, 2020; Kl^oh et al., 2020; Cuevas, 2020; Gomez et al., 2020). The adoption of this combination was done through the use of the disease variables (factors) presented in the SIR model and expressing them in the form of agents (people) who have these variables and carry out their activities within a specific environment (closed environment). This is done so that their movements are monitored and defined in advance by certain rules, to reach the final state of each agent’s characteristics carried out during a certain period of time. The SIR model has been improved and derived into other more comprehensive models, such as Susceptible, Exposed, Infectious and Recovered (SEIR) (Korobeinikov, 2009) which is used in this work.

The validation process in such models depends either on comparing the performance of the developed model with the performance of a proven mathematical model or by comparing it with the actual performance in the real-world environment (Khalil et al., 2012; Alzu’bi, Abu Alasal & Watzlaf, 2020).

In the current work, we simulated the spread of the COVID-19 in Irbid City, North of Jordan, and allocated a specific area to ensure the validity and realism of the results which will be called region X. Two Jordanian nationals who arrived from overseas to attend a wedding event of one of their relatives initiated the spread of the epidemic in a large part of the community. However, due to the strict NPI measures taken by the Jordanian government in Irbid city, the spread of the epidemic across the rest of the community was curbed. The objective of the current research was to evaluate the effect of the strict NPI taken by the government in controlling the spread of the epidemic following the wedding. This includes presenting the situation in Irbid city under more than one scenario.

Further, our simulation presented patch X and the community for males and females in addition to the age distribution of the population (virtually) along with workplaces, markets, a wedding hall, a hospital and an isolation area. We used the SIR model in its SEIR extension because we have taken into consideration the exposed cases that are contagious and not confirmed to be infected but still in the latent period. Details of these contents are provided in the proposed model in “Materials and Methods”.

The results confirmed that the approximate actual situation regarding the number of confirmed infections that result from the wedding event are very close to the predicted number by our model after applying the measures taken to the community of region X. However, we also take into consideration the commitment of the people to abide by or not to abide by the laws and this reflects the tiny difference between the results. By the end of this research, we found that, if preventive measures are not applied, more than 85% of the population might be affected by this pandemic. In addition to the increased number of deaths, and the length of the recovery period.

Materials and Methods

This case study simulates the event that happened in Irbid city, where two infected individuals participated in a wedding ceremony. Thus, we developed an agent-based model to test hypotheses regarding the spread of COVID-19 in a small area of Irbid called X.

SEIR models

The delay between the infection and being infectious (with symptoms) state can be incorporated within the original SIR model by adding an exposed state. SEIR (Korobeinikov, 2009) (Susceptible–Exposed–Infectious–Recovered) model is a type of mathematical pandemic diseases models. In this type of models, individuals experience a long latent period (“exposed” state) and in this situation the individual is infected but not yet infectious Fig. 1. Expresses the SEIR disease states transmission.

Figure 1 SEIR model states transmission.

The infection rate, β, is the rate of spread, which represents the probability of disease transmission between the susceptible and the infectious individual. The latent rate, σ, is the rate of incubation which means the period needed for individuals to become infectious (the average of incubation is 1/σ). The Recovery rate, γ = 1/D, is set by the average duration, D, of infection.

The infection rate, β, is the rate of spread, which represents the probability of the disease transmission between the susceptible and the infectious individual. The latent rate, σ, is the rate of incubation, which means the needed period for individuals to become infectious (the average of incubation is 1/σ). The Recovery rate, γ = 1/D, is set by the average duration, D, of infection.

In a closed population, (no births or deaths from other reasons), SEIR model is defined as:

dSdt=−βSIN

dEdt=βSIN−σE

dIdt=σE−γI

dRdt=γI

Where N (is the total population) = S + E + I + R (Korobeinikov, 2009).

Proposed SEIR model definition

We are using the SEIR epidemic model (Korobeinikov, 2009) as the disease transmission model, all agents are initialized to be in a Susceptible (S) state. Then the pathogen factor is inserted into the community under a specific circumstance and certain persons become in an Exposed (E) state with an infection chance. The exposed state defines the state where an agent is in the latent period of the disease. After a probable time during the incubation, the disease symptoms may appear on the person and his state will move to the Infected (I) state. Some cases may not show symptoms. However, they are still contagious. During the period of infection, symptoms of the disease may increase, complications may occur and the condition may be worse and may cause death. A person is transferred to a Death (D) state under a probability. If the patient’s state of health is managed with high immunity, his body will overcome the disease within a probability and recover, which indicates moving to a full Recovery (R) state. The transformation from one state to another is subjected to potential values from the results of previous research, which will be determined later on this paper. An illustration of the state transition is provided in Fig. 2.

Figure 2 Disease states transformation during infection.

During the movement of agents, the disease can be transmitted between pairs of susceptible-infectious agents that are close to each other on the environment grid. For every agent move, the disease state is checked in addition to its neighbors. The transmission probability is determined by comparing the already specified transmission probability or the infection chance with the immunity value of the agent. If contagion occurs, the status of the agent with a susceptible state is set to be exposed, and at this time, the cycle of the disease begins. The same case is considered when an agent becomes infected. There are two probabilities to be compared, first is the recovery probability and the second is the death probability; in which both of them depend on the existence of high-low immunity level or chronic diseases.

Environment definition

The model community is based on the 1,500-census population of area X located to the south of Irbid city. Agents of this model are placed on a grid that looks like a map that consists of houses, working area divided between shops and building, a wedding hall, a hospital, and an isolation area., the environment is illustrated in Fig. 3 People of this community are engaged in daily activities that reflect the NPI taken by the Jordanian government. The pathogen is inserted inside the community during the wedding ceremony and then it spreads between the population as a consequence of the individual’s interactions and their mobility.

Figure 3 Visual environment.

The initial environment of the simulated area that contains shopping and working area, houses and people who live there, a hospital with medical staff, isolation area, and the wedding hall.

Agent factors: agent characteristics are predetermined and established before the run time. Thus, the first two infected cases, index cases, are initialized based on particular characteristics. These characteristics represent features related to age, sex, occupation, disease status (susceptible, exposed, infected, recovered, and death), dwelling location, household, relatives, and work location.

Building factors: The building types represent the important places that people move through during this situation. Visualized buildings are houses, wedding hall, hospital, and isolation area, shopping and working area. An illustration of the visual environment is provided in Figure 3.

Pandemic disease factors: The model consists of fixed values related to the collected facts about the COVID-19 virus (WHO, 2020; WorldMeters, 2020; Hu et al., 2020).

Latent period: period between the infection by a pathogen and the ability to infect other susceptible persons.

Infection chance: the risk of transmitting the disease from one person to another person.

Recovery chance: a person’s likelihood of recovering once the infection is over.

Infection period: the elapsed period to recover from exposure to a pathogen.

Death rates: a person’s likelihood of transmission to the death state.

Values of these factors are listed in Tables 1 and 2.

Table 1 Related factors to Covid-19 and their definitions associated with the values of each factor.

Factor	Value [1,2,4]	
Latent period	14 days	
Infection period	14 days	
Infection chance	1.4–2.5%	
Recovery chance	97–99.75%	
Note:

The data collected from WHO (2020), WorldMeters (2020), and Hu et al. (2020).

Table 2 The death rate factor distribution related to agent’s ages.

Age	Death rate (%)	
60+ years old	0.085	
50–59 years old	0.04	
10–49 years old	0.02	
0–9 years old	0	
Note:

The data collected from WorldMeters (2020).

Occupation categories: people inside this community have many different occupations distributed based on age and sex. An individual might be a doctor or a nurse as part of a medical staff working inside the hospital. So, they will be on duty. Men and women might be working in the shops or the Governorate offices inside region X. Housewives will stay at home with their children, and their movements will be restricted to only the nearby area around their homes.

Time factors: the model is set-up for 24 h per day, so a series of actions will happen during the day from hour to hour. These actions related to people’s activities inside the community. Hence, there is a schedule between 1 and 24 h that corresponds to successive hours starting from Sunday to Saturday.

Activities factors: everyday, individuals will find daily activities that they must perform. In this environment agents are not active all day’s hours. For example, during the sleeping times and the times of the curfew, everything will be locked down inside the area X. And hence, the model has a set of to-do actions for each defined time slot. These actions define the particular behavior of an agent-based on each individual’s occupation.

Experimental setups of the model

We used the NetLogo software to build the SEIR agent-based model, which is a multi-agent programmable modeling environment (NetLogo, 2020). Our code is published on GitHub (https://bit.ly/3nzDk9t).

The focus of this study is on region X residents who were under curfew and limited movements even within their geographic and administered boundaries. The effectiveness of these measures was assessed and the success rate of these interventions are reported. Scenarios:Herd immunity: area X without internal quarantine where people inside area X are allowed to move inside their community.

NPI: area X is under curfew and lockdown, quarantine. This represents a real time simulation of the strict NPI measures implemented following the wedding ceremony.

Experimental factors of the disease

All values used are based on the daily reports of the Jordanian Ministry of Health (JMH, 2020), world meters website (WorldMeters, 2020), and the World Health Organization (WHO, 2020). Table 1 defines the factors used in the model. Table 2 shows the value of death rates as estimated in JMH (2020).

Results

In the scenario of herd immunity, the results showed a large number of infections within the virtual community of 1,500 residents, where the number of positive cases reached 918, and a total of 96 deaths were reported three weeks after the epidemic started. The period of community recovery was within 47 days. Figure 4 shows the daily updates, this situation was epidemic, as results illustrated in Fig. 5. As for the NPI scenario that considered strict NPI, the results proved the ability of the implemented intervention measures to control the spread of the disease. The number of infections reached 74 cases only, which is much lower than the number of infections in the first scenario. Regarding the number of deaths, the total number was 5 cases. The community recovery period was within 29 days with full commitment. Table 3 shows the statistical results of our model comparing the values of the Herd immunity results with the NPI results.

Figure 4 (A) Agent’s activity factors during time factors with quarantine (B) Agent’s activity factors during time factors without quarantine.

Figure 5 Epidemic impact of no quarantine.

The simulated environment with the impact of no quarantine. Black dots represent normal people, yellow dots represent exposed people, red dots represent infected people and pink dots represent recovered people.

Table 3 The statistical results of our model comparing the two scenarios under study (Herd immunity with the NPI).

Scenario	Infections	Deaths	Social recovery time	
Herd immunity	918 persons	96 persons	47 days	
NPI	74 persons	5 persons	29 days	

The daily updates are shown in Fig. 6. As shown in Fig. 5, and we can notice that the red points indicate the infected people. They are distributed inside the community in separate places, including medical crews inside the hospital. Yellow dots represent people who are in direct contact and at risk of infection. This indicates the seriousness of the situation and the spread of the epidemic within the community. The epidemical localization is illustrated in Fig. 7. The results of this scenario are similar to the situation in Jordan (the real system) during that time, where the number of officially reported COVID-19 cases attributable to the wedding ceremony was 85 cases and the number of deaths was 1 case (Yusef et al., 2020). Figure 8 shows the results in the real system applied in Jordan, only for the first month during the case of Irbid wedding. The convergence in the results between the simulated real time scenario and the actual situation is presented in Fig. 8. The behavior of the curve and the timelines showed good compatibility.

Figure 6 (A) Daily updates with applying quarantine. (B) Daily updates without applying quarantine.

Figure 7 The epidemic localization.

The simulated environment after the end of the disease spreading. The pink dots represent recovered people. In this case, it is easy to detect the epidemic localization.

Figure 8 The daily updates for real situation for the first month in Jordan, and the convergence between the real-word results and the simulated results from the number of infections perspective.

Covid-19 real daily cases during the period from 16 March to 15 April. The red line expresses the divergence of our results as explained in Fig. 6 along with the real daily cases as recorded by the WorldMeters (WorldMeters, 2020).

Ro explains the possibility that an infected person can infect other people. This measure can be determined through several factors such as infectious period, contact rate, and mode of transmission (WHO, 2020). We detected that the Ro value in the case of herd immunity was 5.7, which means that each infected person can infect 5 to 6 people. This explains the large number of infections that happened in our case study since no NPI measures were taken. On the other hand, with the use of NPI measures, the Ro was 2.1, which means that each infected person can infect only two people.

Discussion

Simulation of COVID-19 spread within social gathering from developing countries are scarce, if any. In the current study, we simulated the spread of COVID-19 within a local community following a wedding ceremony from a Jordanian city utilizing two scenarios. The results indicated that the community behavior of strict NPI measures scenario were stable and secured low transmission among residents within the affected area (Fig. 5). In this article, we built a model to represent the wedding event occurring in the Irbid city that spreads infection among the participants. The environment represents the local community within a region X, representing a small part of the Jordanian community. The aim is to measure the effect of strict NPI on controlling the spread of COVID-19. Actual results during the first month of the spread of the infection among the participants in the wedding ceremony is compared with the results of our model. Also, we represented the situation without protective measures. Results indicated the effectiveness of the strict NPI measures in reducing the number of reported cases, deaths, and duration of the epidemic. If no controlling interventions are taken, the outbreak of infections and deaths will happen. Besides, we were able to locate the starting location of the epidemic, which is a basic controlling measure that must be performed. The total number of cases and the duration of the epidemic was dramatically reduced when NPI were considered in a real time simulation as that implemented by the Jordanian government following the wedding. Also, this scenario was able to identify the exact location and the start and end dates of the epidemic.

Strict NPIs at the community level following social gatherings seem to be effective measures to control the spread of COVID- 19 pandemic. This is an evidence-based and data driven message that should be of critical importance for risk communication as it clearly justifies local strict lockdown and its effects on controlling the spread of COVID-19 infection at the local level, and subsequently, the national one. Previous investigations from Jordan has provided evidence that strict national lockdown was effective in reducing the number of national cases and shortening the time of the epidemic, still limited evidence has emerged focusing on sub-national levels or within outbreak settings. Our result, therefore, represents a clear public health message that could be utilized by stakeholders dealing with outbreaks of infectious nature. It indicates that the early on NPI measures could save lives and reduce the utilization of healthcare services even when the economic burden is of concern. Communicating this message to the public may ensure cooperation and adherence to public health measures as the combined efforts of both the government and the public are needed to control the epidemic. The way the public respond to the implemented NPI measures is critical to the presentation of the epidemiological curve of the epidemic. This data-driven approach is a critical public health message to safeguard people’s commitment and to, possibly, aid the efforts of other countries with similar outbreaks.

Considering the potential of future infectious disease epidemics emerging at the local levels, the results under discussion are promising in guiding public health policies aiming at controlling further dissemination of such epidemics. The results of the simulation could be easily transmitted into public health decisions for other infectious diseases that have both asymptomatic and symptomatic phases and both being contagious. This will provide swift decisions and effective implementation of NPI measures. On the other hand, the presented results showed clearly the potential devastating effects of relaxing public health interventions, especially in the absence of a vaccine or antiviral therapies. This may help future research to develop a clear insight that may contribute to implementing public health relevant policies, and to contributing to public health forecasting teams. This especially true as modelling and simulation research is relatively new and underutilized in epidemiology and public health research. Therefore, current research findings broaden our understanding of the possible impact due to potentially overwhelming COVID-19 cases, and allowed better estimation of limitations within available medical resources, such as hospital beds. Date-driven planning and mobilizing of resources will be easier when our results are considered and model results are utilized by public health decision makers.

Hence, the current model showed that the strict NPI measures following the social wedding are vital to contain the epidemic and cannot be relieved. This was also mitigated by local widespread testing and contact tracing, which strongly contributed to a rapid resolution of the epidemic. In the proposed model, however, the effect of such public health measures was not adjusted for.

Conclusions

Results from our proposed model confirm that local NPI measures following social gatherings can potentially reduce the infection peak, given that the diagnosed individuals enter quarantine state and are less likely to infect susceptible individuals, and can shorten the epidemic time, i.e. helping end the epidemic more quickly. As such, it provides support to healthcare settings to be properly function and be able to provide effective, and efficient, care to COVID-19 cases needing medical services. we can conclude that strict NPI measures were successful in reducing the number of infections during the time of the epidemic. This evidence-based public health approach using applied simulation modeling named agent-based modeling should be considered for risk communication and to measure the effectiveness of strict NPI measures.

Additional Information and Declarations

Competing Interests

Author Contributions

Data Availability

The authors declare that they have no competing interests.

Amal Alzu’bi conceived and designed the experiments, analyzed the data, prepared figures and/or tables, authored or reviewed drafts of the paper, and approved the final draft.

Sanaa Abu Alasal performed the experiments, analyzed the data, prepared figures and/or tables, and approved the final draft.

Khalid A. Kheirallah analyzed the data, authored or reviewed drafts of the paper, and approved the final draft.

Valerie Watzlaf analyzed the data, authored or reviewed drafts of the paper, and approved the final draft.

The following information was supplied regarding data availability:

The netlogo implementation is available at GitHub: https://github.com/SanaaAsal/COVID-19-Simulation-Study--The-Effect-of-Strict-Non--Pharmaceutical-Interventions-NPI-on-Controlli

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
