# Peer review of "COVID-19 simulation study—the effect of strict non-pharmaceutical interventions (NPIs) on controlling the spread of COVID-19"

_PeerJ, doi:10.7717/peerj.11172_

## Round 0.1 · original submission · Major Revisions

Based on the reviewers' comments, I think your manuscript has scientific merit to be published in PeerJ. However, there are some issues which you must address in a revised version of the text. Please, see the reviewers' reports in order to have more information.

Best regards,
Dr Palazón-Bru

Reviewer 1 ·

Basic reporting

Line 21: Our life -> our lives

Line 43: at the end of 2019

Line 103: in -> In

Line 145: each day -> everyday; day’s -> daily; he -> they

Line 146-149: improve style

Line 152: Give better reference

Improve the quality of figures.

Better to use pre-symptomatic than asymptomatic.

Provide references of application of agent-based simulation modeling in COVID-19 or similar pandemic situations

Experimental design

Give reference of your code in the manuscript.

Provide discussion on the mathematical model of SEIR and process of its representation in the agent-based simulation model.

Line 147-49 "The model has a set of to-do actions for each defined time slot. These actions define the particular behavior of an agent-based on each individual’s occupation."

Please illustrate line 147 - 149 through some figure or table.

Validity of the findings

Provide statistical analysis to validate your results on daily updates of simulated results and real results.

Additional comments

Please address the comments.

Regards,

Reviewer 2 ·

Basic reporting

This manuscript evaluated the effect of NPI measures on a social event in a developing country setting using SIR simulation and compared epidemic curve characteristics utilizing two major scenarios. The manuscript also utilized a public health approach to present the results but also conserved the modeling of data. This interdisciplinary approach is much needed to present findings where public health decision makers can provide evidence-based decisions when dealing with COVID-19 in developing countries. As well, event specific modeling is scarce for COVID-19 and little is known beyond national level modeling. Overall, the manuscript presentation was appropriate using the before mentioned clarifications
It seems more appropriate to use NPI measures instead of NPIs. This goes for the whole manuscript.
Introduction: SARS-CoV-2 instead of SARS-COV-2.
Also, severe acute respiratory syndrome coronavirus 2 (SARS-CoV-2) should be used at the first statement of SARS-CoV-2.
Line 46: Nearly 20 million.
Line 68: 1927as
Line 75: effectiveness?
Line 78: SEIR, SIQR,
Line 108: in an Irbid city, please delete “an”.
Line 241: effectiveness of the strict NPI measures in reducing the number of reported cases, deaths, and duration of the epidemic.

Experimental design

Line 146: what are the probabilities used for moving between stages? They are critical for the SEIR models to run. Values presented in table 1 are not comprehensive. For example, what is the probability of death used? Contact rate (Beta)? Alpha? I see that death rates were presented in table two but are better merged with table 1 and a reference be provided, if available.
What is the name of the software used to perform the analysis? A simple description will be an added value to this model.
Legends for figures need to be added as most do not have such.

Validity of the findings

The model based approach used in this study is unique. But, does it have any limitations? Are there limitations for this study as seen by the authors?

Reviewer 3 ·

Basic reporting

no comment

Experimental design

no comment

Validity of the findings

no comment

Additional comments

--
Review comments for manuscript ID: “54026-v0”, entitled “COVID-19 simulation study - The effect of strict nonpharmaceutical Interventions (NPIs) on controlling the spread
of COVID-19” of journal “PeerJ”.

--
General:
Alzu’bi et al. employed an agent-based epidemiological modelling analysis to study the effects of strict Non-Pharmaceutical Interventions (NPIs) on controlling the spread
of COVID-19 pandemic during a wedding ceremony from Irbid, Northern Jordan with consideration of Two infected nationals of Jordan, who arrived in Jordan about a week before the event, initiated the spread of the pandemic within the contact community.
We observed the deterioration of the state of society and the spread of the epidemic among individuals in the absence of preventive measures. Also, our results show that the herd immunity case was an epidemic, with a high level of spread among the community with 918 cases during a short interval of time. On the other hand, the preventive measures scenario shows a totally controlled spread with only 74 cases applied on the same interval of time. Furthermore, a convergence in the actual results of the real system with the hypothetical system were detected in the case of applying the strict NPI measures. Strict NPIs at the community level following social gatherings seem to be effective measures to control the spread of the COVID- 19 pandemic. The analyzing outcomes support the main results in this work with biologically and clinically reasonable settings. Generally, I am fine with their results, and I also think this study is interesting and worth publishing (after revision round). Aside from the interesting parts of the work, I have some comments and suggestions for the authors.





--
Comments

--I suggest the abstract should be rewritten. Since it is not clear whether the authors are reporting an event or using a case scenario to highlight the effects of NPIs on controlling the spread of the COVID-19 pandemic.

--Each abbreviated term should be defined in its first introduction throughout the manuscript. E.g. severe acute respiratory syndrome coronavirus 2 (SAR-CoV-2).

--“,” should be inserted between a name and year for each citation. Such as (Cuevas 2020) should be (Cuevas, 2020). I suggest to double check with reference format of PeerJ.

--Epidemiological background should be updated to include up-to-date information on COVID-19 in Jordan and worldwide.

--In line 64-65, the sentence “Simulation can be represented utilizing mathematical equations.” Should be made clear and precise.

--I wonder what the authors are trying to report in line 74-75?

--Line 77-79, the authors should write citation from academic archives not just Wikipedia. There are numerous studies to cite. The sentence should also be made clear and precise. Also, the typo “)” should be corrected.

--“(Khalil et al. 2012); (Alzu’bi et al. 2020).” Should be written as “(Khalil et al., 2012; Alzu’bi et al., 2020).” Similar issues should be corrected throughout the manuscript.

--Line 143, “1 and 24” should be “1 and 24 hours”

--In Figure 2, what the right and left boxes means with regard to the current scenario? I wonder where they came from? And what they represent in the model with respect to the infection of the disease? i.e., Infection chance, event, asymptomatic, …

--The first sentence in the conclusion section is a bit narrow, I suggest the authors to improve this part.

--I wonder why the authors numbered the subsection, while the sections were not numbered? These should be made consistent throughout.

--I noticed that the results are based on the data not on the model? Any explanation?

--I appreciate the authors have done a very nice and comprehensive discussion mainly focusing on the technical part. It would be more appreciated if the author could elaborate more from the epidemiology and public health sides, which could fit more to the journal.

-- I suggest some important published papers should be discussed or cited. Such as:
.. https://doi.org/10.1038/s41579-020-00459-7
.. https://idpjournal.biomedcentral.com/articles/10.1186/s40249-020-00718-y
.. https://www.nejm.org/doi/full/10.1056/NEJMOa2001316
.. https://doi.org/10.1016/j.ijid.2020.01.050
.. https://doi.org/10.1016/j.ijid.2020.02.058
.. https://doi.org/10.1016/j.mbs.2020.108364

---

## Round 0.2 · Major Revisions

Still pending some changes suggested by two of the reviewers, which you should address in a new revised version of the text.

Reviewer 1 ·

Basic reporting

no comments

Experimental design

More explanation of agent-based modelling and the process of representing SEIR into agent-based modelling is needed.

Without mathematical formulation difficult to place the parameters of the model.

Validity of the findings

Difficult to understand the parameter values of the model.

Additional comments

Dear Authors,

Thanks for your efforts.
Please address the comments to improve the readability of the article.

Thanks

Reviewer 2 ·

Basic reporting

Clear English
Clear figures and tables
Clear results

Experimental design

Well defined scope and methods

Validity of the findings

Results and conclusions are well stated

Additional comments

I believe they did all the suggested changes. so, I recommend to accept the paper.

Reviewer 3 ·

Basic reporting

no comment

Experimental design

no comment

Validity of the findings

no comment

Additional comments

General:

I appreciate the authors have carefully addressed all the issues raised by the reviewers that are useful to improve this manuscript. I think the paper is recommended for publication after consideration of the minor issue below.

--
Comments

The authors should clearly state their findings in the conclusion section.

---

## Round 0.3 · Minor Revisions

Still pending some minor modifications suggested by the reviewers.

Reviewer 1 ·

Basic reporting

Line 30-31: “the herd immunity case was an epidemic”, please use better way to express this.
Line 306: better to use pre-symptomatic in place of asymptomatic
Figure 3, 5 and 7: Provide legends for different colors of dots.
Figure 8: Provide legends

Experimental design

Infection chance and recovery chance hasn’t been shown in the mathematical formulation in line 134 to 138.
The values of parameter such as (exposed initially) E(0) and (recovered initially) R(0) is not mentioned.

Validity of the findings

The results section may include the value of the parameters such as R naught for herd immunity case and NPI case.

Additional comments

Dear Authors,

Thanks for incorporating comments.
Please do the needful as per new comments.

Regards,

Reviewer 3 ·

Basic reporting

no comment

Experimental design

no comment

Validity of the findings

no comment

Additional comments

The authors have addressed all the comments.
I think the manuscript can be accepted after some minor correction below.

Equation 3 of the SEIR system should be dI/dt...
that is
dS/dt=-βSI/N
dE/dt=βSI/N-σE
dI/dt=σE-γI
dR/dt=γI.

---

## Round 0.4 · accepted · Accept

All the reviewers' concerns have been correctly addressed. Therefore, I am pleased to make this acceptance decision.

Reviewer 1 ·

Basic reporting

No comments

Experimental design

No comments

Validity of the findings

No comments

Additional comments

Dear Authors,

Thanks for incorporating comments.

Regards,